# Rethinking Positional Embedding: A Case Study in Temporal Event Sequence Modelling

**Effat Farhana**
Department of Computer Science
Vanderbilt University
Nashville, TN 37235, USA
`effat.farhana@vanderbilt.edu`

## Abstract

In this paper, we present a time-decaying encoding as an alternative to sinusoidal positional encoding in the transformer architecture. We evaluate our approach in the context of an educational domain involving 14,043 question-solving interactions from 1,260 students. We argue that including time-based attention can be beneficial for event sequence modeling applications where the inter-event time intervals and the interpretation of the model's prediction both are crucial.

## 1 Introduction

Transformer architectures need positional embedding. In this paper, we ask: do we always need positional embedding? We present an empirical analysis of a weighted temporal event sequence modeling to encode timestamps as an alternative to positional embedding. We argue that temporal information can be more valuable to capture sequential events in the health and education sector.

## 2 Background and Related Work

Researchers have used time-decaying attention (Ghosh et al., 2020), inter-event embedding (Gu, 2021) instead of positional encoding. We present our study in the context of deep knowledge tracing (DKT)–an approach to predict a student's correctness in question-solving based on the student's past question-solving records (Piech et al., 2015; Pandey & Karypis, 2019; Pandey & Srivastava, 2020; Ghosh et al., 2020). In the education domain, the *forgetting curve theory* states that the students' memory decay exponentially with time (Ebbinghaus, 2013). Thus, researchers incorporated forgetting behavior into DKT: elapsed time between the consecutive events (Shin et al., 2021; Nagatani et al., 2019), and elapsed time between events with similar skill tagging (Nagatani et al., 2019).

## 3 Methodology

We present our model architecture below. The study context is in the Appendix.

**Text Embedding.** We use the universal sentence encoder USE (Cer et al., 2018) to encode action text into a $d = 512$ vector, $\mathbf{E} \in \mathbb{R}^d$

**Action Type Embedding.** We learn an action embedding size of $d = 512$ for four actions.

**Score Encoding.** We extend the score of each question to a vector size $d$.

**Response Encoding.** We compute the cosine similarity between questions and responses and extend the value to a vector size $d$.

We concatenate these four feature vectors to form an input for each timestamp $i$, $\mathbf{x_i} \in \mathbb{R}^{4d}$.

**Time Relation.** This component takes into account students' forgetting behavior. We took the negative exponential of the elapsed time $\Delta_i$ between the $T_{th}$ action timestamp, $t_T$ and $i_{th}$ previous action's timestamp, $t_i$ as: $\mathbf{R_T} = [exp(-\Delta_1), exp(-\Delta_2), ..., exp(-\Delta_{T-1})]$

**Question Relation.** This component computes the cosine similarities between the text embedding of $T_{th}$ question, $\mathbf{E_T}$ and a previous question at $t_i$, $\mathbf{E_i}$. We denote this component as $\mathbf{R_Q}$.

Table 1: Mean of 5-fold CV on test fold. Params. = Parameter Count, Mem. = Memory in MB

| Method | Sinusoidal Positioning Embedding | | | | Time Decaying Relation | | | |
|---|---|---|---|---|---|---|---|---|
| | AUC | Accuracy | Params. | Mem. | AUC | Accuracy | Params. | Mem. |
| $Full$ | 0.85 | 0.79 | 11763728 | 11.22 | 0.80 | 0.76 | 11558928 | 11.02 |
| $Q^{rem}$ | 0.72 | 0.74 | 11763728 | 11.22 | 0.73 | 0.72 | 11558928 | 11.02 |

**Attention.** We compute the dot product attention $\alpha_{i,j}$ (Vaswani et al., 2017) for a $\mathbf{q_i}$ query vector of a student's question attempted at time $i$. We combine $\alpha_{i,j}$ with the **Time** and **Question** relation, $\mathbf{R}$ as: $\beta_{i,j} = \lambda\alpha_{i,j} + (1 - \lambda)\mathbf{R_j}$. Here, $R_j$ is the $j_{th}$ coefficient of $\mathbf{R}$ and $\lambda$ is a trainable parameter. The final attention is $\mathbf{Att_i} = \sum_{j<i}\beta_{i,j}.\hat{\mathbf{x}}_\mathbf{j}\mathbf{W^V}$, $\mathbf{v_j}$ = value vectors (Details in Appendix).

## 4 EXPERIMENT DESIGN AND RESULT

Table 1 presents results of two forms of encodings: sinusoidal position encoding (Vaswani et al., 2017) and exponential time decaying relation. We present two sets of experiments for each encoding. The $Full$ denotes the full model and $Q^{rem}$ denotes excluding the question relation from the full model. We report five-fold cross-validation results. The task is a binary classification task. Hyperparameter details and dataset statistics are in the Appendix.

Table 2: The Student's Action Sequence of Figure 1

| Time | Action Text (Score) |
|---|---|
| T1 | In your own words, explain how the arches were formed. Use some of the science terms we used in the past. *(Score = 0)* |
| T2 | Explain the meaning of physical change. What does it affect? *(Score = 0)* |
| T3 | Explain the meaning of physical change. What does it affect? *(Score: True = 0, Predicted = 0)* |

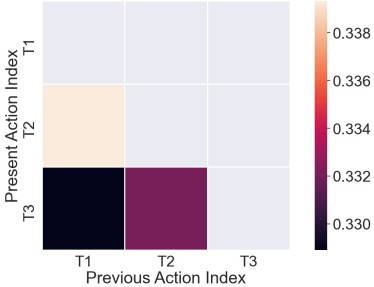

Figure 1: Attention Weight Averaged on Four Heads: Full Model (Test Fold)

**Result.** The sinusoidal encoding $Full$ model outperforms the time-decaying relation $Full$ model. In both encodings, removing the $Q^{rem}$ component from the $Full$ model results in a performance drop. Our model's parameter size and memory are relatively smaller. From Table 1, we observe the model's size and other components both contribute to performance.

Figure 1 shows the attention weight assigned by our model for one student. Table 2 shows question texts and scores. We observe the model puts more weight on "T2" as a time-decaying result in predicting the question score at "T3". The question at "T3" is a resubmission of "T2". The student scored zero on both questions at "T1" and "T2". The model predicted zero at "T3".

## 5 DISCUSSION AND CONCLUSIONS

Incorporating time and model interpretation in the sequential prediction is important in the health sector, (e.g., disease progression modeling (Zhang, 2019)) and education domain (See Section 2). However, the sinusoidal positional encoding does not capture the inter-event time intervals. In this paper, we present an alternative encoding that puts decayed weight on distant events. Our proposed approach's strength is *education theory-informed design* that incorporates (i) students' forgetting behavior and (ii) provides a rationale for the model's prediction. Although the sinusoidal model outperforms ours, we are not looking for the best performance. Rather, we advocate for an alternative of positional encoding with *comparable performance* and domain application specificity.

URM STATEMENT

"The author acknowledges that the sole author of this work meets the URM criteria of ICLR 2023 Tiny Papers Track."

ACKNOWLEDGEMENTS

We acknowledge Dr. Teomara Rutherford and Dr. Collin F. Lynch for their feedback on this project.

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

## A  APPENDIX

**Study Context and Dataset Statistics.** Our dataset is from a K-12 science reading platform, Actively Learn (AL). AL is a popular online reading platform used in US Schools. We identified four types of actions *within* the AL system: question attempts, annotating, highlighting, and vocabulary lookup.

The dataset statistics are: the number of assignments = 378, number of questions = 1,260, number of students = 1,680, and number of interactions = 14,043. The mean, standard deviation, and median of the sequence are 6.4, 3.6, and 6.

**Experiment Settings.** We report five-fold cross-validation results. The task is a binary classification task.

1. **Full Model.**
   (a) Time Decaying Relation: Proposed architecture.
   (b) Sinusoidal Positioning Embedding. This model uses the sinusoidal position embedding Vaswani et al. (2017) and does not include the time-decaying relation.
2. $\mathbf{Q^{rem}}$. Excluding the **Question Relation** component from the model architecture.

**Hyperparameters.** We used a batch size of 200, embedding size of 512, 300 epochs, learning rate = $1e^{-3}$, number of attention heads = 4, and early stopping if the validation AUC did not increase over five epochs.

**Attention:** $\mathbf{W^Q}, \mathbf{W^K}$, and $\mathbf{W^V}$ are projection matrices for query, key, and value space, respectively with dimension $\mathbb{R}^{d*d}$.

$$\alpha_{i,j} = \text{Softmax}(\frac{\mathbf{q_i}\mathbf{W^Q}.(\mathbf{k_j}\mathbf{W^K})^{\mathbf{T}}}{d}), \text{all} j < i$$
$$\mathbf{Att_i} = \sum_{j<i} \alpha_{i,j}.\mathbf{v_j}\mathbf{W^V}$$

(1)

$\mathbf{k_j}$ and $\mathbf{v_j}$ denote respectively key and value vectors for previous actions, $j < i$. In DKT models, keys and values are past events, so $j < i$.

$\mathbf{R} = \text{Softmax}(\mathbf{R_T} + \mathbf{R_Q})$.

**Source Code:** Link to source code.

