# OpenReview forum: "RETHINKING POSITIONAL EMBEDDING: A CASE STUDY IN TEMPORAL EVENT SEQUENCE MODELLING"
_ICLR.cc/2023/TinyPapers — Submitted to Tiny Papers @ ICLR 2023_

### Official Review · Reviewer_XiyW · 2023-03-28

**Confidence:** 3

**Summary Of Contributions:**

This paper provides an empirical evaluation of the transformer architecture for a binary classification task using a real-world education-related dataset. Two encodings have been compared, namely, sinusoidal positional encoding and time-decaying encoding. The authors recall particular attention to time-decaying encoding in the context of health and education data.

**Rating:**

Clear, Correct, and Reproducible (CCR): a submission which meets the reviewing criteria

**Strengths And Weaknesses:**

Strengths: The paper provides all details on the implementation of the transformer method and the used encodings.

Weaknesses: The trade-off between performance (measure with AUC/accuracy) and Memory is not very promising.

**Suggested Changes:**

- I suggest consistently using "quotes" for T1 and T2;
- I recommend answering the following questions: Are there any limitations that the authors identified? Rather than only comparing performance and memory consumption, are there other aspects that time-decaying encoding would be a better alternative?;
- The last phrase seems incompleted (This paper presents an empirical analysis of alternative encoding.). Alternative encoding for what?

Minor comments:
- Please correct the word "Including", which is capitalized in the abstract;
- Similarly, the word "Thus" is capitalized in Section 2 within the phrase "[...] with time Ebbinghaus (2013), Thus [...]".

---

> ### Author Response · Authors · 2023-05-24
> **Addressed reviewer's comment by adding more clarification.**
>
> 1. "I suggest consistently using "quotes" for T1 and T2;"
>
> Response: Done. Thanks.
>
> 2. "I recommend answering the following questions: Are there any limitations that the authors identified? Rather than only comparing performance and memory consumption,..."
>
> Response: Applications where the event ordering and time intervals both are important, such as sepsis prediction and student’s score prediction based on historical records (knowledge tracing).
>
> 3. "The last phrase seems incompleted (This paper presents an empirical analysis..."
> Response: Revised Discussion and conclusion section.
>
> Minor Comments:
>  Addressed. Thanks.

---

### Official Review · Reviewer_Zzyj · 2023-04-01

**Confidence:** 4

**Summary Of Contributions:**

The paper presents empirical analysis of temporal event sequence modeling to encode positional information in a Transformer architecture. The paper also explores an alternative for sinusoidal positional embedding in Transformer architecture.

**Rating:**

Clear, Correct, and Reproducible (CCR): a submission which meets the reviewing criteria

**Strengths And Weaknesses:**

**Strengths**

- The paper is clear and concise.
- The paper provides the details of their architecture, training data statistics, and hyperparameters used which is a good step toward reproducibility.
- The authors did a thought study and applied their approach to specific domains(healthcare & education).
- The paper follows ICLR paper requirements.

**Weaknesses**

- The reported experiments are good but don’t highlight the impacts of the time-decaying encoding approach.
- No code is provided to support the conducted experiments.

**Suggested Changes:**

- The paper is written well and clearly but it would be great to conduct more experiments. An example could be to show how their model works without positional information vs sinusoidal encoding vs time-decaying encoding.
- It is unclear how much model size plays with encoding type. Did the time-relation perform worse than sinusoidal embedding because the model used for time-relation is smaller than later? Conducting more experiments would perhaps clarify your findings.
- The conclusion(and discussion) is concise but it would benefit from having your major findings and how they connect with your research question.

---

> ### Author Response · Authors · 2023-05-24
> **Added more experiments and elaborated results.**
>
> Suggested Changes by Reviewer and Response
>
> 1. "The paper is written well and clearly but it would be great to conduct more experiments. ..."
>
> Response: Added more experiment variations. Please refer to Table 1.
>
> 2. "It is unclear how much model size plays with encoding type..."
> Response: Model size and other components contribute to the model's performance as presented in Table 1.
>
> 3. "The conclusion(and discussion) is concise but it would benefit ..."
> Response: Elaborated conclusion and discussion connecting with our research question.
>
> 4. Added link to source code in the Appendix.

---

### Meta-Review · Area_Chair_wFWG · 2023-04-05

**Recommendation:** Invite to revise
**Confidence:** 4

**Metareview:**

This paper presents an empirical comparison of the transformer’s sinusoidal positional encoding and time-decaying encoding a time series application of the education domain involving 14,043 question-solving interactions from 1,260 students. The paper provides detailed implementations. Reviewers were concerned about the lack of analysis to show (i) the impacts of the time-decaying encoding approach, (ii) how much model size plays with encoding type, (iii) the pros and cons of the proposed method compared with existing techniques.




**Summary:**

This paper provides an empirical evaluation of the transformer architecture for a binary classification task using a real-world education-related dataset. The paper provides clear implementation details but the experiments are somewhat preliminary and more in-depth studies are encouraged.

**Comments And Feedback To The Authors:**

There are some writing issues that should be fixed. Please refer to the detailed comments of the reviewers, along with a more in-depth proofreading. Also, please handling the citations correctly when the references are as part of the text or not.

**Reason For Not Giving A Higher Recommendation:**

The experiments do not well support the goal of "rethinking the positional embedding". The authors may want to clearly present the pros and cons of different alternatives with more thorough analysis and provide valuable suggestions for chosing the optimal ones in different scenarios.

**Reason For Not Giving A Lower Recommendation:**

N/A

---

### Decision · Program_Chairs · 2023-04-07

**Decision:**

Invite to archive

**Comment:**

Although reviewers considered this submission CCR, the meta-reviewer and reviewers identified areas for improvement. Please address all questions and revise the manuscript to make improvements.